# Achievement of the low-density lipoprotein cholesterol goal among patients with dyslipidemia in South Korea

**Siin Kim**[1], **Sola Han**[1], **Pratik P. Rane**[2], **Yi Qian**[2], **Zhongyun Zhao**[2], **Hae Sun Suh**[1]*

**1** College of Pharmacy, Pusan National University, Busan, Korea (South), **2** Amgen Inc., Thousand Oaks, CA, United States of America

* haesun.suh@pusan.ac.kr

## Abstract

### Background

It is important to achieve the low-density lipoprotein cholesterol (LDL-C) goal recommended by clinical guidelines in managing the risk of cardiovascular (CV) events, however, the current management of LDL-C in actual clinical settings is suboptimal. We examined the LDL-C level among patients with dyslipidemia against the 2015 Korean guidelines, the crude rates of CV events based on LDL-C goal achievement, and the factors associated with LDL-C goal achievement.

### Methods

This was a retrospective cohort study using the National Health Insurance Service–National Health Screening Cohort (NHIS-HEALS) database from 2006 to 2013. Patients who had a health examination with LDL-C measurement between January 1, 2007, and December 31, 2011 were identified. Patients were required to have at least one diagnosis of dyslipidemia during the 1 year before the index date, defined as the first date of LDL-C measurement. The 2015 Korean guidelines were used to measure LDL-C goal achievement based on the CV risk level. Crude CV event rates were calculated for total and individual CV events as the number of events divided by person-years (PYs) during the follow-up period. CV events included acute coronary syndrome, ischemic stroke, peripheral artery disease, CV death, and all-cause death. Factors associated with LDL-C goal achievement were assessed using logistic regression.

### Results

In the NHIS-HEALS database, 69,942 patients met the eligibility criteria: 36.7%, 22.5%, 20.1%, and 20.6% were among the very high-, high-, moderate-, and low-risk groups for the CV events, respectively, as defined by the 2015 Korean guidelines. Approximately half of the patients with dyslipidemia (47.6%) achieved their recommended LDL-C goal, but the achievement rates were substantially different across CV risk levels (17.6%, 47.2%, 66.9%, and 82.4% for very high-, high-, moderate-, and low-risk groups, respectively; *P*<0.0001).

**Data Availability Statement:** The datasets used and analyzed during the current study are not publicly available. There are legal or ethical restrictions on sharing this data publicly. Data for these analyses were made available to the authors

through National Health Insurance Service (NHIS), through a formal application process (https://nhiss. nhis.or.kr/bd/ab/bdaba021eng.do). Researchers can access the NHIS data only when they meet the all of following conditions: (i) Korean citizenship; (ii) Institutional Review Board approval; and (iii) Permission by NHIS data provision review committee (contact number: +82-33-736-2431). Because NHIS strictly prohibits researchers from sharing the raw data (i.e., individual-level data which is not summarized) to unauthorized persons who are not involved in the study, other researchers who are interested in using NHIS database should request the data through a formal application process. For patient confidentiality reasons, public data sharing is restricted even if the data is anonymized.

**Funding:** This research was funded by Amgen, Inc.; URLs to sponsor's website: https://www. amgen.com/. Qian, Rane are employees of Amgen Inc and own stocks in the company. Zhao was an employee of Amgen Inc at the time this research was performed. The funder provided support in the form of salaries for authors PPR, YQ, and ZZ, but the funder did not have any additional role in the study design, data collection and analysis, decision to publish, or preparation of the manuscript. The specific roles of these authors are articulated in the 'author contributions' section (Qian, Rane and Zhao contributed in development of study design, analysis, decision to publish and in preparation of the manuscript).

**Competing interests:** I have read the journal's policy and the authors of this manuscript have have the following competing interests: Qian, Rane are employees of Amgen Inc and own stocks in the company. Zhao was an employee of Amgen Inc at the time this research was performed. Suh, S Kim, and Han received research grants from National Research Foundation, Ministry of Health and Welfare, Ministry of Food and Drug Safety, Korea Health Industry Development Institute, Abbvie Korea, Amgen Inc, Amgen Korea, Handok-Teva, Ipsen Korea, and Pfizer Korea. This does not alter our adherence to PLOS ONE policies on sharing data and materials.

The crude event rate of total CV events during the follow-up period in the LDL-C goal non-achievers was higher than that in the LDL-C goal achievers (24.35/100 PYs vs. 11.93/100 PYs; *P*<0.0001). LDL-C goal achievement was significantly associated with patient characteristics, including age, sex, body mass index, lipid-modifying therapy, and CV risk level.

## Conclusions

In South Korea, LDL-C goal achievement among patients with very high or high CV risk was suboptimal. Patients who did not achieve the goal showed a higher rate of CV events during the follow-up period than patients who achieved the goal. LDL-C management strategies should be highlighted in dyslipidemia patients who are less likely to achieve the goal, such as female, overweight or obese patients, patients not adherent to statin, or patients with very high or high CV risk.

## Introduction

Low-density lipoprotein cholesterol (LDL-C) has been well established as a causal factor of cardiovascular disease (CVD), and therefore, lowering LDL-C through lipid-lowering therapy is among the primary means of lowering risk of cardiovascular (CV) events [1–5]. In South Korea, the 2015 Korean guidelines for the management of dyslipidemia suggested LDL-C goals based on an individual's CV risk level, similar to the National Cholesterol Education Program (NCEP) Adult Treatment Panel III (ATP III) guidelines [5, 6].

Previous studies showed that an achievement of recommended LDL-C goal is associated with decreased risk of future CV events among patients with established CVD [7, 8]. Despite the benefit of LDL-C goal achievement, the current management of LDL-C in actual clinical settings is suboptimal, leading to the elevated risk of CV events among patients with dyslipidemia [9, 10]. Unfortunately, there is a paucity of evidence for current status of LDL-C goal achievement among broad population with various CV risk levels in South Korea.

The achievement of the recommended LDL-C goal can be affected by patients' demographic and clinical characteristics [11–14]. Especially patients at higher risk of CV event or using less potent statin therapy are less likely to achieve their LDL-C goals [12, 13].

The objectives of this study were to examine (1) the LDL-C goal achievement among patients with dyslipidemia in Korea, (2) the rates of CV events based on LDL-C goal achievement, and (3) the factors associated with LDL-C goal achievement by using a nationwide health screening data.

## Patients and methods

### Study design and data source

A retrospective cohort study using the National Health Insurance Service–National Health Screening Cohort (NHIS-HEALS; NHIS-2017-2-292) database from January 1, 2006, to December 31, 2013 was conducted. As a single insurer, the National Health Insurance covers the entire Korean population, which consists of 97.2% of health insurance enrollees and 2.8% of medical aid beneficiaries [15]. In Korea, the National Health Insurance supports general health screening tests every two years for beneficiaries with workplace insurance and yearly for beneficiaries who own a business. The NHIS-HEALS includes information on a cohort of 514,866 enrolled health screening participants, randomly selected among health screening

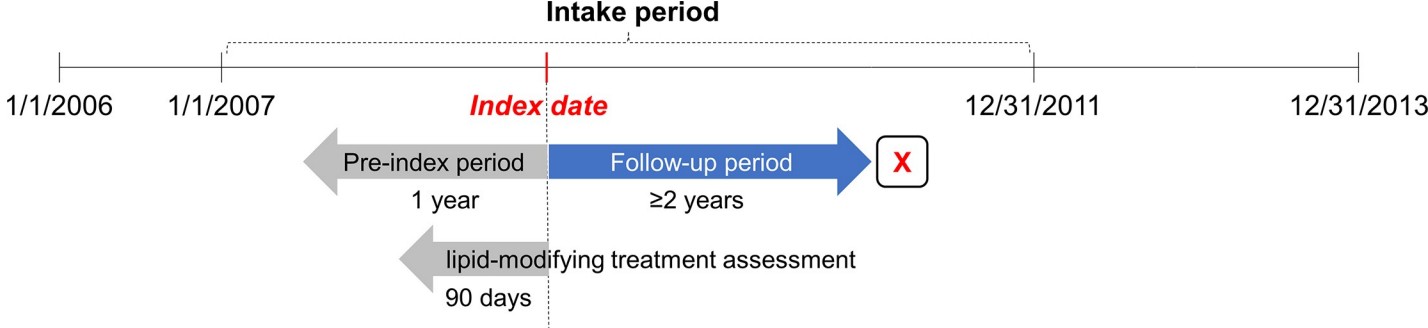

**Fig 1. Study design scheme.** The index date was defined as the first date of health examination with low-density lipoprotein cholesterol measurement.

participants who were aged between 40 and 79 years in 2002 and 2003, and followed up until 2013. Medical aid beneficiaries (i.e., low-income families) were excluded in this study, because they have begun to receive general health screening tests supported by NHIS since 2012. The database includes socioeconomic variables, such as residence, the date of death, cause of death, and income level, and details of medical treatment and health examination [16]. This study was exempt from the Institutional Review Board review of Pusan National University (PNU IRB/2016_136_HR).

## Study population

We identified patients who underwent health examination with LDL-C measurement between January 1, 2007, and December 31, 2011, because LDL-C measurements were available from 2007 in the database (Fig 1). The index date was defined as the first date of health examination with LDL-C measurement. The patients were required to have at least one diagnosis of dyslipidemia (International Classification of Diseases, Tenth Revision, [ICD-10] code of E78) during the 1 year before the index date. The patients were excluded from the study if they had any missing values in major risk factors as defined by the Korean guidelines, such as smoking, blood pressure, high-density lipoprotein cholesterol (HDL-C), and family history of coronary artery disease (CAD).

We used the definition of LDL-C goals per CV risk level recommended by the 2015 Korean guidelines for the management of dyslipidemia as follows: (1) <70 mg/dL for patients with very high risk, (2) <100 mg/dL for patients with high risk, (3) <130 mg/dL for patients with moderate risk, and (4) <160 mg/dL for patients with low risk. For each patient, CV risk level was assessed during the 1 year before the index date.

## Subgroup

Based on the 2015 Korean guidelines, the very high-risk group was defined by a history of CVD (angina [both stable and unstable], myocardial infarction, ischemic stroke, transient ischemic attack, peripheral artery disease, and revascularization), and the high-risk group was defined by a history of risk factors equivalent to CAD (abdominal aneurysm and diabetes mellitus). Diagnosis and procedure codes to identify very high- and high-risk groups are listed in S1 Table [17, 18]. Among patients without history of CVD or risk factors equivalent to CAD, patients with two or more major risk factors were defined as moderate-risk patients, whereas patients with zero or one risk factor were defined as low-risk patients. Major risk factors included the following conditions: (1) current smoking, (2) hypertension defined as blood pressure BP ≥140/90 mmHg or taking antihypertensive, (3) HDL-C <40 mg/dL, (4) family

history of premature CAD (male aged <55 years and female aged <65 years), and (5) male aged ≥45 years and female aged ≥55 years. If a patient had HDL-C ≥60 mg/dL, one factor was excluded from the total number of risk factors [6]. In the NHIS-HEALS data, information on family history of CAD is collected via questionnaire, but whether the CAD is premature is unclear. Therefore, we regarded patients as having a risk factor if they had family history of CAD regardless of its prematurity.

## Medication adherence and LMT classification

The use of lipid-modifying treatment (LMT) was assessed based on the proportion of days covered (PDC), during the 90 days before the index date (Fig 1) [19]. PDC was defined as the sum of days' supply of medication divided by the number of days in the observation period (90 days in this analysis). The patients were regarded as being compliant to LMT if their PDC was >80% [19, 20]. The patients were classified into the following four groups based on PDC: (1) "no LMT" group without previous prescriptions for all of LMT, (2) "adherent (statin)" group with a PDC of ≥80% for statin regardless of PDC for non-statin, (3) "adherent (non-statin)" group with a PDC of ≥80% for non-statin, such as ezetimibe and fibrate while having a PDC of <80% for statin, and (4) "non-adherent" group with a PDC of <80% for all LMTs. If a patient was taking more than one type of LMT, PDC was calculated separately for each type of LMT.

## Outcome measures

For each patient, LDL-C goal achievement was assessed by comparing the LDL-C value on the index date with the LDL-C goal defined based on the 2015 Korean guidelines. The proportion of patients who achieved the LDL-C goal was explored for total patients, CV risk groups, and LMT groups.

To calculate crude CV event rates, all patients were followed up from the index date until the occurrence of a CV event, death, or the end of the study period (December 31, 2013), whichever came first. CV events included acute coronary syndrome (myocardial infarction and unstable angina), ischemic stroke, peripheral artery disease, CV death, and all-cause death. Diagnosis codes to identify CV events are listed in S2 Table.

We examined the factors associated with LDL-C goal achievement with variables, such as age, sex, CV risk level, LMT group, Charlson comorbidity index (CCI) score, HDL-C, and smoking status [21].

For all outcome measures, we conducted a sensitivity analysis by defining LMT groups based on the PDC calculated during 1 year before the index date. In addition, we assessed LDL-C goal achievement per CV risk level recommended by the NCEP ATP III guidelines, which had been used in clinical practices before the 2015 Korean guidelines were developed. Although both the 2015 Korean guidelines and NCEP ATP III guidelines suggested LDL-C goals based on the CV risk groups, there are some differences in definitions for the CV risk groups and recommended LDL-C goals. The NCEP ATP III guidelines recommended <100 mg/dL for both very high- and high-risk patients, while an optional goal of <70 mg/dL could be considered for very high risk patients [5]. In the sensitivity analysis, we did not consider the optional goal for very high-risk patients.

## Statistical analysis

The categorical variables were presented as frequencies with proportions, and continuous variables were presented as means with standard deviations. Crude CV event rates were calculated as the number of CV events divided by person-years (PYs) during the follow-up period. We

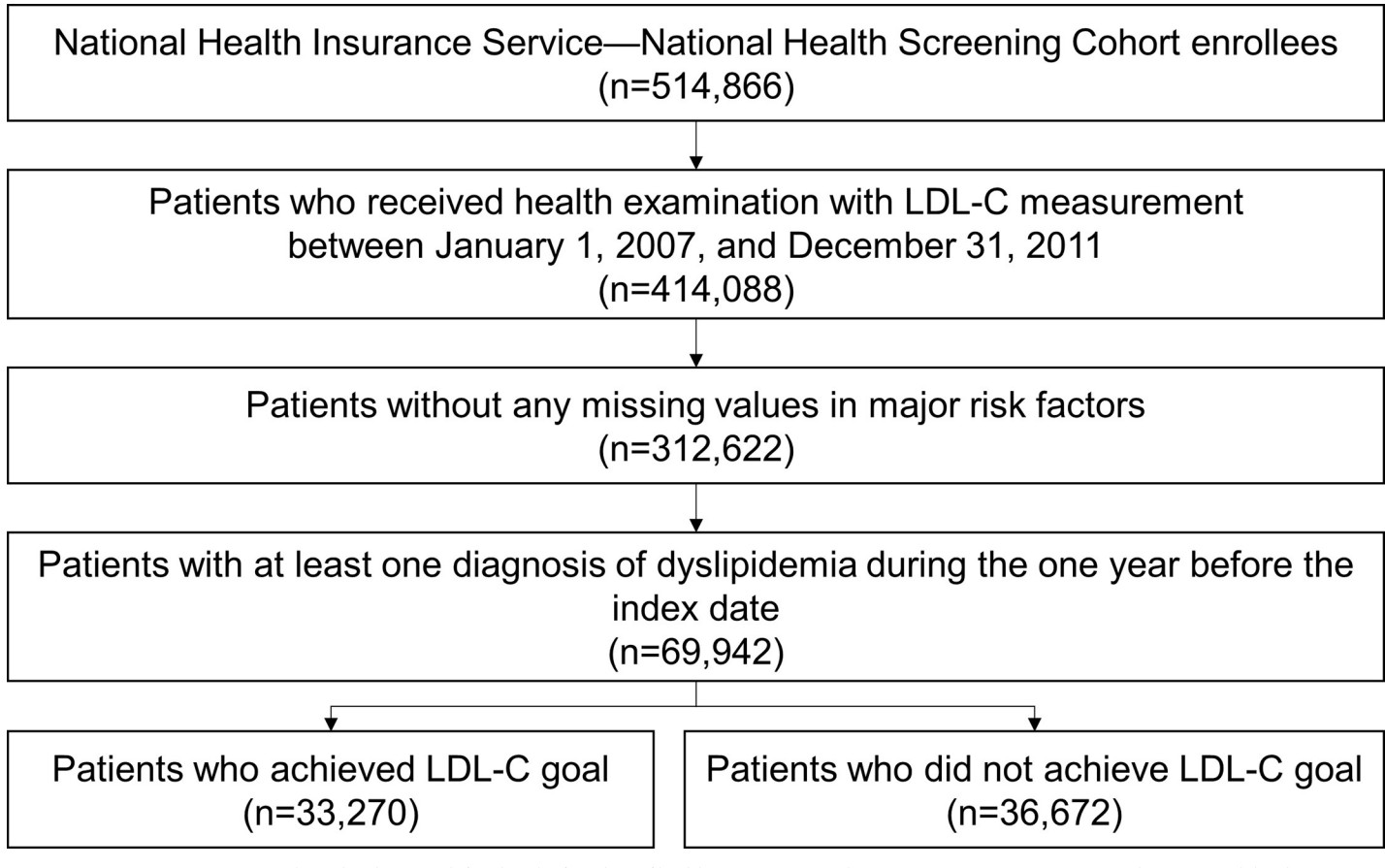

**Fig 2. Sample selection process.** The index date was defined as the first date of health examination with LDL-C measurement. LDL-C goals per CV risk level were defined by the 2015 Korean guidelines. LDL-C, low-density lipoprotein cholesterol.

performed a logistic regression and estimated odds ratios (ORs) with 95% confidence intervals (CIs) to examine the factors associated with LDL-C goal achievement. Patients without body mass index (BMI) data on the index date were excluded from the logistic regression. SAS software (version 9.4 for Windows; SAS Institute, Cary, NC, USA) was used for statistical analysis. $P<0.05$ was considered statistically significant.

## Results

### Patients' characteristics

We included 69,942 patients who were eligible for the study (Fig 2). The mean age was 61.1 years, and 51.3% were female. Approximately half of patients were not taking LMT, and only a third of patients were taking statin with a PDC of ≥80%. Based on the 2015 Korean guidelines, 36.7%, 22.5%, 20.1%, and 20.6% patients had very high, high, moderate, and low risk for future CV events, respectively (Table 1). Among these patients, 33,270 (47.6%) achieved the LDL-C goal and 36,672 (52.4%) did not achieve the LDL-C goal, and statistically significant differences were found in the baseline characteristics between LDL-C goal achievers and non-achievers. LDL-C goal non-achievers were on average older (61.9 years vs. 60.2 years) and had more comorbidities (CCI score, 2.7 vs. 1.9), higher proportion of women (52.8% vs. 49.6%), and higher CV risk levels (very high-risk group, 57.7% vs. 13.6%) compared with LDL-C goal achievers.

**Table 1. Baseline characteristics of patients with dyslipidemia.**

| Characteristics | LDL-C goal achievers (n = 33,270) | LDL-C goal non-achievers (n = 36,672) | P-value[a] |
|---|---|---|---|
| **Age, years** | 60.2 ± 8.3 | 61.9 ± 8.4 | <0.0001 |
| **Female** | 16,488 (49.6%) | 19,380 (52.8%) | <0.0001 |
| **LDL-C, mg/dL** | 89.9 ± 29.9 | 129.3 ± 40.8 | <0.0001 |
| **HDL-C, mg/dL** | 55.5 ± 32.9 | 53.9 ± 32.6 | <0.0001 |
| **Systolic BP, mmHg** | 126.3 ± 14.8 | 127.7 ± 15.4 | <0.0001 |
| **Diastolic BP, mmHg** | 77.5 ± 9.7 | 78.1 ± 9.9 | <0.0001 |
| **Body mass index[b], kg/m²** | 24.6 ± 3.0 | 24.9 ± 3.0 | <0.0001 |
| Underweight (<18.5 kg/m²) | 480 (1.4%) | 406 (1.1%) | |
| Normal (18.5–22.9 kg/m²) | 9,524 (28.6%) | 9,337 (25.5%) | |
| Overweight (23–24.9 kg/m²) | 9,084 (27.3%) | 10,274 (28.0%) | |
| Obese (25–34.9 kg/m²) | 14,062 (42.3%) | 16,533 (45.1%) | |
| Severely obese (≥35 kg/m²) | 120 (0.4%) | 122 (0.3%) | |
| **Family history of heart disease** | | | <0.0001 |
| Yes | 2,499 (7.5%) | 3,330 (9.1%) | |
| No | 30,730 (92.4%) | 33,271 (90.7%) | |
| Unknown | 41 (0.1%) | 71 (0.2%) | |
| **Smoking status** | | | <0.0001 |
| Never | 21,893 (65.8%) | 24,839 (67.7%) | |
| Previous smoker | 6,856 (20.6%) | 6,976 (19.0%) | |
| Current smoker | 4,521 (13.6%) | 4,857 (13.2%) | |
| **Comorbidities** | | | |
| Myocardial infarction | 364 (1.1%) | 872 (2.4%) | <0.0001 |
| Ischemic stroke | 899 (2.7%) | 3,838 (10.5%) | <0.0001 |
| Peripheral artery disease | 2,032 (6.1%) | 11,164 (30.4%) | <0.0001 |
| Diabetes mellitus | 9,875 (29.7%) | 17,319 (47.2%) | <0.0001 |
| Chronic renal failure | 327 (1.0%) | 394 (1.1%) | 0.23 |
| **Charlson comorbidity index** | 1.9 ± 1.7 | 2.7 ± 2.0 | <0.0001 |
| **CV risk level** | | | <0.0001 |
| Very high risk | 4,533 (13.6%) | 21,164 (57.7%) | |
| High risk | 7,448 (22.4%) | 8,320 (22.7%) | |
| Moderate risk | 9,411 (28.3%) | 4,652 (12.7%) | |
| Low risk | 11,878 (35.7%) | 2,536 (6.9%) | |
| **LMT group[c]** | | | <0.0001 |
| No LMT | 13,712 (41.2%) | 20,401 (55.6%) | |
| Adherent (statin) | 13,514 (40.6%) | 9,521 (26.0%) | |
| Adherent (non-statin) | 463 (1.4%) | 721 (2.0%) | |
| Non-adherent (all LMT)[c] | 5,581 (16.8%) | 6,029 (16.4%) | |

Values are presented with either mean ± SD or number (%).

BP, blood pressure; CV, cardiovascular; HDL-C, high-density lipoprotein cholesterol; LDL-C, low-density lipoprotein cholesterol; LMT, lipid-modifying treatment; PDC, proportion of days covered; SD, standard deviation.

[a]Continuous values were compared using Student $t$ test. Categorical values were compared using $\chi^2$ test and Fisher's exact test.

[b]Body mass index was evaluable in 33,265 of LDL-C goal achievers and 36,651 of LDL-C goal non-achievers.

[c]LMT groups were defined based on PDC: (1) "no LMT" without previous prescriptions for all of LMT, (2) "adherent (statin)" with a PDC of ≥80% for statin regardless of PDC for non-statin, (3) "adherent (non-statin)" with a PDC of ≥80% for non-statin while having a PDC of <80% for statin, and (4) "non-adherent" with a PDC of <80% for all LMTs.

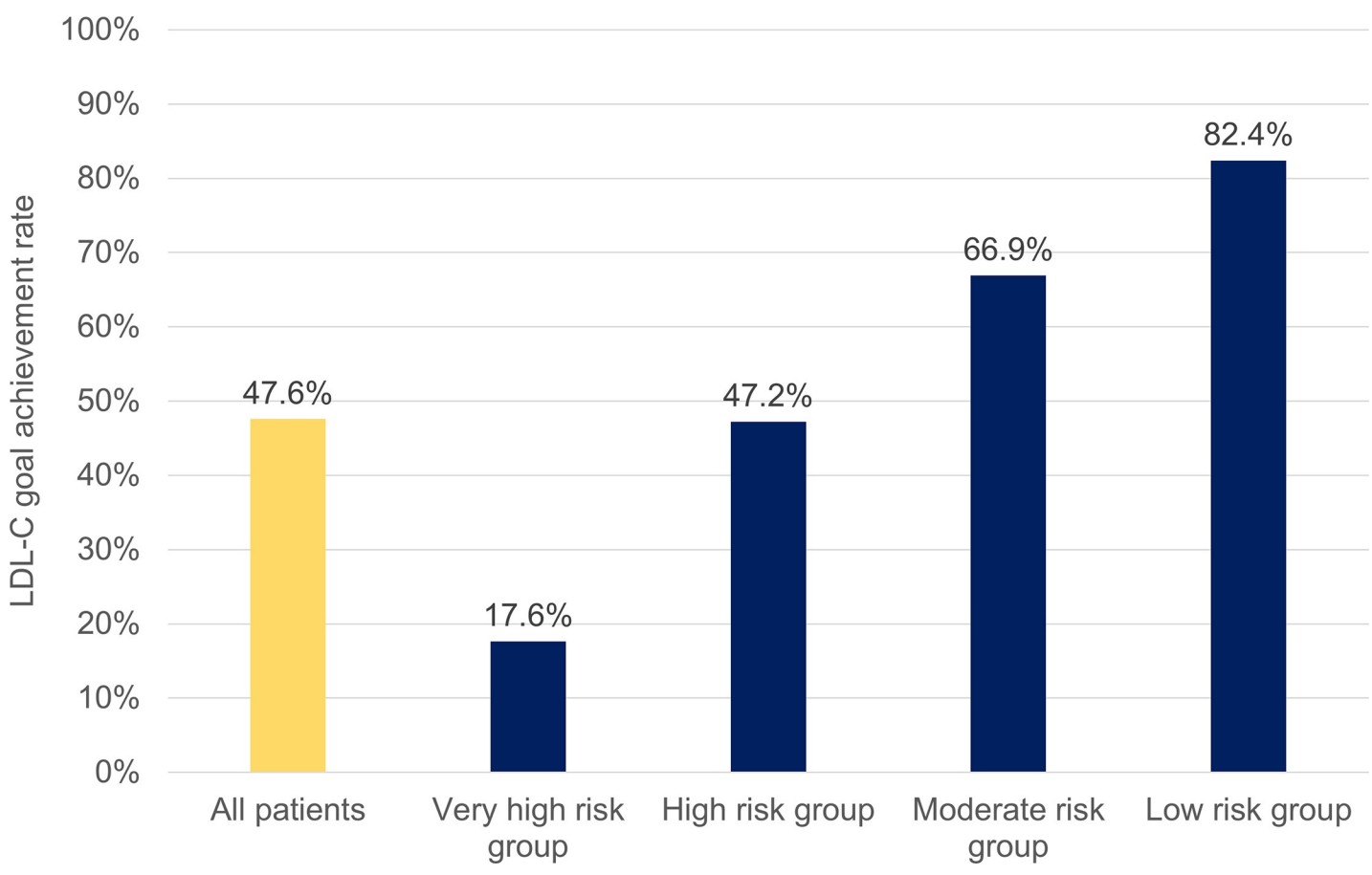

**Fig 3. LDL-C goal achievement rates among patients with dyslipidemia (based on the 2015 Korean guidelines).** LDL-C, low-density lipoprotein cholesterol.

## LDL-C goal achievement

Among patients with dyslipidemia, 47.6% achieved their recommended LDL-C goal, but the achievement rates were substantially different based on CV risk levels (Fig 3). Only 17.6% of patients with very high-risk level achieved the LDL-C goal (<70 mg/dL), whereas 82.4% of patients with low-risk level achieved the LDL-C goal (<160 mg/dL).

Fig 4 shows LDL-C goal achievement rates and average LDL-C levels categorized by the use of LMT. LDL-C goal achievement rates were different with the use of LMT. The patients who were adherent to statin tended to have lower LDL-C levels, and therefore presented higher rates of goal achievement compared with other LMT groups (i.e., no LMT, adherent to non-statin, and non-adherent). The proportion of patients adherent to statin was 36.3%, 16.7%, 8.0%, and 3.3% in very high-, high-, moderate-, and low-risk group, respectively. However, even among the patients adherent to statin, 70.6%, 26.2%, 9.6%, and 2.8% in very high-, high-, moderate-, and low-risk group, respectively, still needed further reduction in their LDL-C levels to meet the recommended LDL-C goals.

## CV event rates based on LDL-C goal achievement

The crude rate of total CV events, including all-cause death, ischemic stroke, acute coronary syndrome, and peripheral artery disease, during the follow-up period in LDL-C goal non-achievers (24.35/100 PYs) was higher than that in LDL-C goal achievers (11.93/100 PYs) (Table 2). Similar trends were identified for the individual CV events. Especially for ischemic

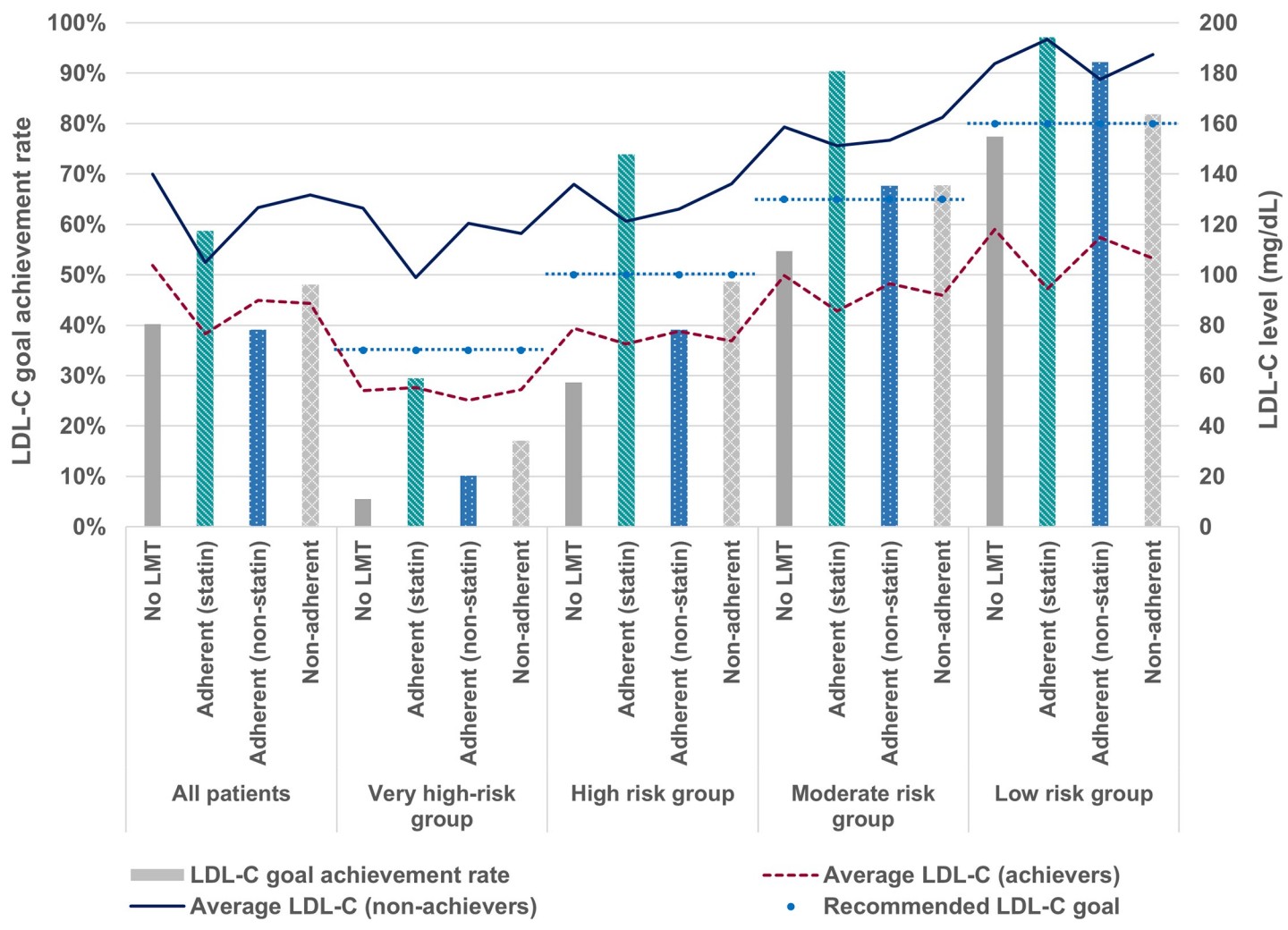

**Fig 4. LDL-C goal achievement rates based on the 2015 Korean guidelines and LDL-C distributions categorized by the use of lipid-modifying treatment.** LDL-C, low-density lipoprotein cholesterol; LMT, lipid-modifying treatment.

**Table 2. Crude cardiovascular event rates based on LDL-C goal achievement.**

| CV events | LDL-C goal achievers | | LDL-C goal non-achievers | | P-value[a] |
|---|---|---|---|---|---|
| | Number of events | Rates per 100 PYs | Number of events | Rates per 100 PYs | |
| Total CV events[b] | 11,560 | 11.93 | 19,890 | 24.35 | <0.0001 |
| All-cause death | 539 | 0.56 | 718 | 0.88 | <0.0001 |
| CV death | 39 | 0.04 | 73 | 0.09 | <0.0001 |
| Acute coronary syndrome[c] | 1,764 | 1.82 | 3,021 | 3.70 | <0.0001 |
| Ischemic stroke | 1,686 | 1.74 | 3,584 | 4.39 | <0.0001 |
| Peripheral artery disease | 7,571 | 7.81 | 12,567 | 15.38 | <0.0001 |

CV, cardiovascular; LDL-C, low-density lipoprotein cholesterol; PY, person-year.

[a]P-values for differences between rates of LDL-C goal achievers and non-achievers.

[b]Total CV events included all-cause death, acute coronary syndrome, ischemic stroke, and peripheral artery disease.

[c]Acute coronary syndrome is a composite of myocardial infarction and unstable angina.

stroke and acute coronary syndrome, the event rates of LDL-C goal non-achievers were 2 to 2.5 times higher than the event rates of LDL-C goal achievers. Peripheral artery disease showed the highest event rates among individual CV events.

### Factors associated with LDL-C goal achievement

Age (OR = 1.003; 95% CI [1.001–1.006]), female (OR = 0.55; 95% CI [0.52–0.58]), LMT group, CV risk level, BMI, CCI (OR = 1.07; 95% CI [1.06–1.09]), and HDL-C (OR = 0.999; 95% CI [0.999–1.000]) showed significant association with LDL-C goal achievement (Fig 5). Compared with patients adherent to statin, other LMT groups (i.e., no LMT, adherent to non-statin, and non-adherent) were less likely to achieve the goals (OR = 0.14; 95% CI [0.13–0.15], OR = 0.23; 95% CI [0.20–0.27], and OR = 0.32; 95% CI [0.30–0.34], respectively). For CV risk levels, patients with higher CV risk were less likely to achieve the goals compared with low-risk group (OR = 0.01; 95% CI [0.01–0.02], OR = 0.09; 95% CI [0.09–0.10], and OR = 0.30; 95% CI [0.28–0.31] for very high, high, and moderate risk, respectively). Compared with patients with normal weight, underweight patients were more likely to achieve the goals (OR = 1.53; 95% CI [1.28–1.82]) while overweight or obese patients were less likely to achieve the goals (OR = 0.84;

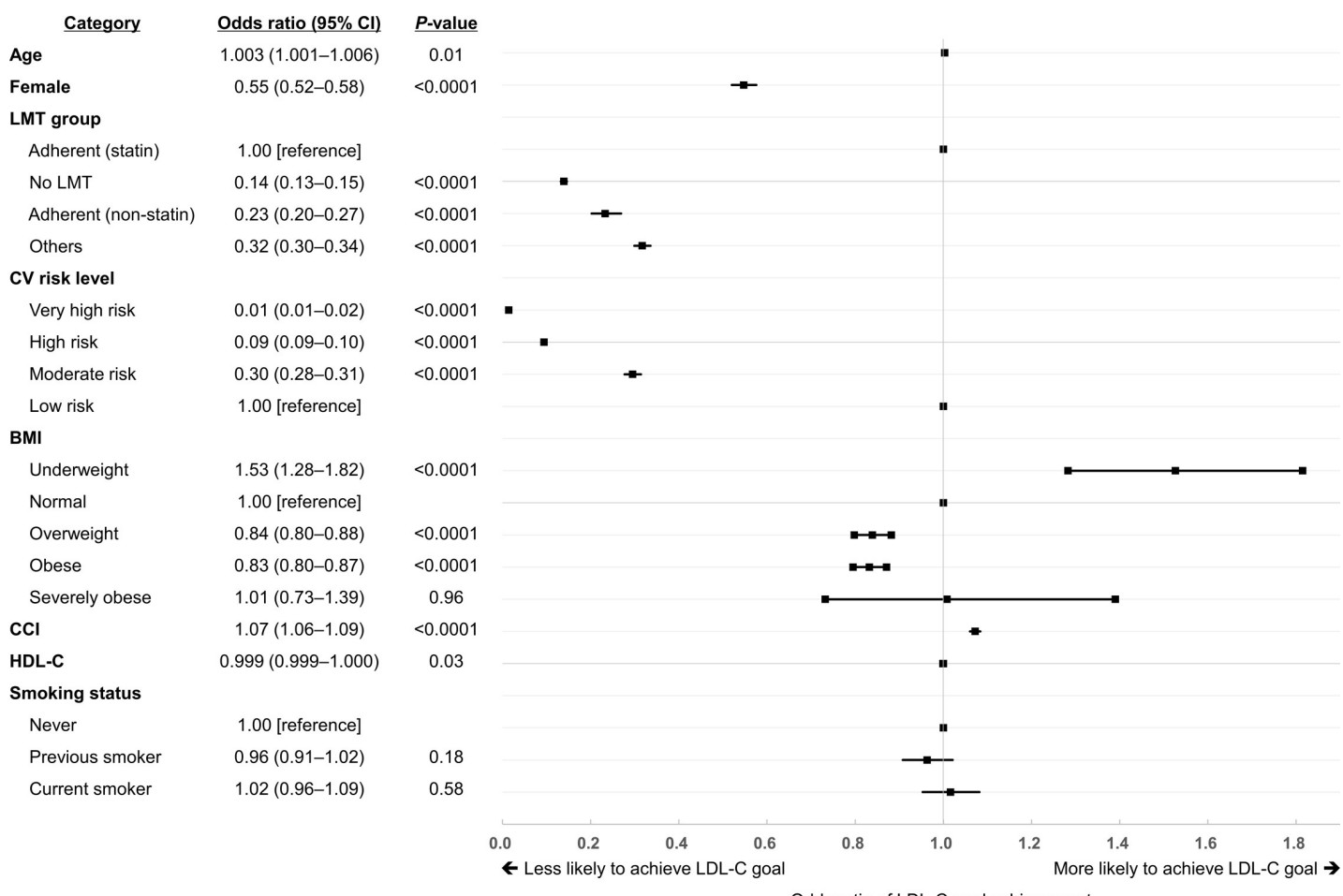

**Fig 5. Factors associated with LDL-C goal achievement.** Twenty-six patients without BMI data on the index date were excluded from the logistic regression. BMI, body mass index; CCI, Charlson comorbidity index; CI, confidence interval; CV, cardiovascular; HDL-C, high-density lipoprotein cholesterol; LMT, lipid-modifying treatment.

95% CI [0.80–0.88] and OR = 0.83; 95% CI [0.80–0.87] for overweight and obese, respectively). Both previous (OR = 0.96; 95% CI [0.91–1.02]) and current smokers (OR = 1.02; 95% CI [0.96–1.09]) did not show significant ORs compared to patients who never smoked.

## Sensitivity analysis

When calculating PDC during 1 year before the index date, the goal achievement rates and the results of logistic regression were similar to those in the analysis using PDC during the 90 days before the index date.

When we defined LDL-C goals based on the NCEP ATP III guidelines instead of the 2015 Korean guidelines, 41,447 (59.3%) patients achieved the LDL-C goal (S1 Fig). Among the very high-risk group, 12,710 (49.5%) patients achieved the LDL-C goal of <100 mg/dL. Other risk groups showed the same LDL-C goal achievement rates when compared with the achievement rates using the 2015 Korean guidelines.

## Discussion

In this retrospective cohort study, which used nationwide health screening data, we demonstrated that less than half of patients with dyslipidemia achieved the LDL-C goal recommended by the 2015 Korean guidelines. The achievement rate was much lower in patients with very high or high risk for future CV events. LDL-C goal non-achievers showed higher rates of CV events during the follow-up period compared with LDL-C goal achievers. Patients' age, sex, LMT, CV risk level, BMI, disease severity, and HDL-C were associated with LDL-C goal achievement.

Because lipid profiles and LDL-C lowering effects are different between Western and Asian population, we used the 2015 Korean guidelines which considered evidences applicable to Koreans [6, 22]. The 2015 Korean guidelines defined a very high-risk group as patients with established CVD and a recommended goal of <70 mg/dL, whereas the NCEP ATP III guidelines defined the very high-risk group as patients with established CVD plus one or more risk factors and a recommended goal of <100 mg/dL (an optional goal of <70 mg/dL) [5, 6]. This means that the 2015 Korean guidelines suggest a stricter LDL-C goal for less severe patients classified as the very high-risk group compared with the NCEP ATP III guidelines.

The Return on Expenditure Achieved for Lipid Therapy in Asia (REALITY-Asia) study examined LDL-C goal achievement and factors associated with goal achievement based on the NCEP ATP III guidelines in six Asian countries [13]. For Korean patients, 45.3%, 28.8%, 17.6%, and 8.3% were classified as very high-, high-, moderate-, and low-risk patients, respectively. The proportion of very high- and high-risk patients in the REALITY-Asia study was higher than that in our study. This might be because the REALITY-Asia study used hospital-based data, whereas our study used national data of a general health screening program that involved a healthier population than hospital-based data. The achievement rate for Korean patients with very high and high risk was 35% in the REALITY-Asia study, with an LDL-C goal of <100 mg/dL. This result could be comparable to our result, where the achievement rates were 49.5% (with an LDL-C goal of <100 mg/dL) and 47.2% (with an LDL-C goal of <100 mg/dL) for very high- and high-risk patients using the NCEP ATP III guidelines, respectively (S1 Fig). Results of sensitivity analysis in the REALITY-Asia study, where 12% of very high- and high-risk patients achieved the goal of LDL-C <70 mg/dL, were also comparable to our result, where 17.6% or very high-risk patients achieved the LDL-C goal of <70 mg/dL. The REALITY-Asia study also showed that age, potency of initial statin treatment, CV risk level, and baseline LDL-C were associated with LDL-C goal achievement. ORs of age (OR = 1.02; 95% CI [1.02–1.21]), very high- and high-risk level (OR = 0.09; 95% CI [0.05–0.13]), and

moderate-risk level (OR = 0.29; 95% CI [0.18–0.46]) were comparable with our results. The Reality China survey also examined LDL-C goal achievement and factors associated with goal achievement among the Chinese population based on the NCEP ATP III guidelines [12]. This study reported lower proportion of very high-risk patients (16%) and higher proportion of high-risk patients (43%) than our study, and the achievement rates were generally low for all risk groups (25.8%). This difference could result from the different definition of CV risk levels and LDL-C goals, and discrepancies in clinical settings between countries. Using three real-world data sources, Jones et al. examined LDL-C goal achievement among high-risk patients treated with statin monotherapy in the United States [23]. The achievement rates were 66.6%–76.8% and 20.1%–26.0% with the LDL-C goal of <100 mg/dL and <70 mg/dL, respectively. In the sensitivity analysis of this study, we applied an LDL-C goal of <100 mg/dL to very high- and high-risk patients based on the NCEP ATP III guidelines, which was adopted from 2003 to 2014 in Korea. The LDL-C goal achievement rate in the very high-risk group has greatly increased in the sensitivity analysis (49.5%), where the NCEP ATP III guidelines were applied, compared with the analysis (17.6%) where the 2015 Korean guidelines were applied.

Compared with other risk groups, the very high-risk group presented a particularly low rate of LDL-C goal achievement. The mean LDL-C of the very high-risk group was 104.1 mg/dL, and 49.5% were at LDL-C <100 mg/dL. Among the LDL-C goal non-achievers in the very high-risk group, patients adherent to statin had a lower LDL-C level compared with the other LMT groups. However, these patients still need to reduce their LDL-C level to reach the goal of <70 mg/dL. Similarly, LDL-C goal non-achievers in the high-risk group also need to further reduce their LDL-C level to reach the goal of <100 mg/dL. This suggests unmet therapeutic needs in lowering LDL-C levels in patients with very high or high risk for CV events. Considering that CV events are related with considerable socioeconomic burden, LMT should be intensified in patients with very high or high risk [24, 25].

To the best of our knowledge, this is the first study of LDL-C goal achievement status among a broad population with various CV risk levels in South Korea. We used a nationwide health screening data over a 8-year period, which was suitable to studying non-communicable disease by providing information on laboratory results, health behaviors, medical treatment, death, and demographics [16]. By using this data, we were able to study LDL-C goal achievement among broad population with various CV risk levels and assess the association between patients' characteristics and LDL-C goal achievement. Moreover, we found that sensitivity analyses provided results generally comparable to the main analysis, assuring the robustness of the main analysis.

This study has several limitations. First, medical aid beneficiaries were excluded in this study because health screening tests for medical aid beneficiaries have been supported by the NHIS since 2012. Second, the family history of CAD was assessed regardless of the time of onset when calculating major risk factors, because the time of CAD onset of a family member cannot be identified in the NHIS-HEALS data. This might misclassify some patients with low risk into the moderate-risk group, leading to underestimated rates of goal achievement for the moderate-risk group. Third, because we defined dyslipidemia and CV events based on diagnosis and procedure codes, some of the events could be misidentified. However, the validity of these codes is partly assured because we defined them based on validation studies conducted in Korea [17, 18]. Fourth, our study period precedes the development and announcement of the 2015 Korean guidelines. Therefore, our data were generated from the previous clinical settings, where LDL-C goals were recommended in line with the NCEP ATP III guidelines. Considering that we analyzed LDL-C goal achievements based on the 2015 Korean guidelines, which had stricter LDL-C goal criteria than the NCEP ATP III guidelines, this may contribute to the low rate of LDL-C goal achievement among the very high-risk patients in this study.

Because we applied the current criteria rather than the criteria corresponding to the study period, this may be a limitation of the study and attention should be paid to interpreting the results. Nevertheless, we used the 2015 Korean guidelines for the purpose of understanding the status of LDL-C goal achievement in accordance with current practices in clinical decision making. Fifth, CV event rates were calculated without adjustment for patients' characteristics, and thus the results should be interpreted with caution. Sixth, we did not consider previous LDL-C levels, which might affect LDL-C goal achievement. It was not feasible for us to consider the LDL-C levels before patients have initiated LMT, because in most cases the health screening tests are conducted biannually in South Korea, resulting in at least a 2-year gap between LDL-C measurements. Instead of using the previous LDL-C level at least 2 years ago, we have considered the recent use of LMT with 90 days (and conducted sensitivity analyses using 6 months and 1 year) before the LDL-C measurement to assess the goal achievement.

Therefore, further studies are needed to explore LDL-C goal achievement with consideration for treatment patterns and baseline LDL-C levels. In addition, the assessment of the risk of CV events based on the LDL-C goal achievement is needed.

## Conclusion

In South Korea, a high proportion of dyslipidemia patients with very high or high CV risk did not meet the LDL-C goal recommended by the 2015 Korean guidelines. The rate of CV events during the follow-up period was higher in patients who did not achieve the goal than in patients who achieved the goal. Although patients adherent to statin tended to have lower LDL-C levels, there was still a need for additional reduction in LDL-C levels to reach the goal. Patients were less likely to achieve the LDL-C goal if they were female, overweight or obese, taking LMT other than statin, and with higher CV risk. More intensive LDL-C management, including regular LDL-C measurement and optimized lipid-modifying therapy, should be highlighted in patients who are less likely to achieve the LDL-C goal, such as female, overweight or obese patients, patients not adherent to statin, or patients with very high or high CV risk.

## Supporting information

**S1 Table. Diagnosis and procedure codes to identify very high- and high-risk groups.**
(DOCX)

**S2 Table. Diagnosis codes to define cardiovascular events.**
(DOCX)

**S3 Table. LDL-C goal achievement rates based on the 2015 Korean guidelines and LDL-C distributions categorized by the use of lipid-modifying treatment.**
(DOCX)

**S1 Fig. LDL-C goal achievement rates among patients with dyslipidemia based on the National Cholesterol Education Program Adult Treatment Panel III guidelines.**
(TIF)

## Acknowledgments

This study used National Health Insurance Service–Health Screening data (NHIS-2017-2-292) made by National Health Insurance Service (NHIS). The authors declare no conflict of interest with NHIS.

## Author Contributions

**Conceptualization:** Siin Kim, Sola Han, Pratik P. Rane, Yi Qian, Zhongyun Zhao, Hae Sun Suh.

**Data curation:** Siin Kim, Sola Han.

**Formal analysis:** Siin Kim, Sola Han, Hae Sun Suh.

**Funding acquisition:** Hae Sun Suh.

**Investigation:** Siin Kim, Sola Han, Pratik P. Rane, Hae Sun Suh.

**Methodology:** Siin Kim, Sola Han, Pratik P. Rane, Yi Qian, Zhongyun Zhao, Hae Sun Suh.

**Project administration:** Siin Kim, Sola Han, Pratik P. Rane, Zhongyun Zhao, Hae Sun Suh.

**Resources:** Sola Han.

**Supervision:** Pratik P. Rane, Yi Qian, Hae Sun Suh.

**Validation:** Siin Kim, Sola Han, Hae Sun Suh.

**Writing – original draft:** Siin Kim, Hae Sun Suh.

**Writing – review & editing:** Siin Kim, Sola Han, Pratik P. Rane, Yi Qian, Zhongyun Zhao, Hae Sun Suh.

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
