## [Decision Letter · Decision Letter 0]

11 Nov 2019

PONE-D-19-24724

Achievement of the low-density lipoprotein cholesterol goal among patients with dyslipidemia in South Korea

PLOS ONE

Dear Hae Sun Suh,

Thank you for submitting your manuscript to PLOS ONE. After careful consideration, we feel that it has merit but does not fully meet PLOS ONE’s publication criteria as it currently stands. Therefore, we invite you to submit a revised version of the manuscript that addresses the points raised during the review process.

We would appreciate receiving your revised manuscript by Dec 26 2019 11:59PM. To enhance the reproducibility of your results, we recommend that if applicable you deposit your laboratory protocols in protocols.io, where a protocol can be assigned its own identifier (DOI) such that it can be cited independently in the future. For instructions see: http://journals.plos.org/plosone/s/submission-guidelines#loc-laboratory-protocols

We look forward to receiving your revised manuscript.

Kind regards,

Luigina Guasti

Academic Editor

PLOS ONE

Journal Requirements:

'I have read the journal’s policy and the authors of this manuscript have the following competing interests: [Qian, Rane are employees of Amgen Inc and own stocks in the company. Zhao was an employee of Amgen Inc at the time this research was performed. Suh, S Kim, and Han received research grants from National Research Foundation, Ministry of Health and Welfare, Ministry of Food and Drug Safety, Korea Health Industry Development Institute, Abbvie Korea, Amgen Inc, Amgen Korea, Handok-Teva, Ipsen Korea, and Pfizer Korea.]'  

We note that one or more of the authors are employed by a commercial company: Amgen Inc

Additional Editor Comments (if provided):

Reviewers' comments:

Reviewer's Responses to Questions

**Comments to the Author**

1. Is the manuscript technically sound, and do the data support the conclusions?

Reviewer #1: Yes

Reviewer #2: Yes

2. Has the statistical analysis been performed appropriately and rigorously? 

Reviewer #1: Yes

Reviewer #2: Yes

3. Have the authors made all data underlying the findings in their manuscript fully available?

Reviewer #1: Yes

Reviewer #2: Yes

4. Is the manuscript presented in an intelligible fashion and written in standard English?

Reviewer #1: Yes

Reviewer #2: Yes

5. Review Comments to the Author

Reviewer #1: This was a retrospective cohort study in a specific ethnic group.

The work raises some questions that should be answered and discussed

What is the rate of CV events divided between LDL goal achievers and not-achievers considering the CV risk class?

Cardiovascular events may have been influenced by failure to achieve targets in other CV risk factors (diabetes, hypertension). Do you have data on this?

No data on BMI?

Are there data on chronic renal failure?

Reviewer #2: In this retrospective cohort study the authors used data from a South Korean National Health Insurance DataBase (2006 to 2013) to assess the LDL-C goal achievement among patients with very high or high CV risk treated with statins and other lipid lowering drugs. The 2015 Korean guidelines were used to measure LDL-C goal achievement based on the CV risk level. 69,942 patients were evaluated. About 50% of the patients with dyslipidemia achieved their recommended LDL-C goal, but the achievement rates were substantially different across CV risk levels (17.6%, 47.2%, 66.9%, and 82.4% for very high-, high-, moderate-, and low-risk groups, respectively; P<0.0001). The crude event rate of total CV events during the follow-up period in the LDL-C goal non-achievers was higher than that in the LDL-C goal achievers (24.35/100 PYs vs. 11.93/100 PYs; P<0.0001). LDL-C goal achievement was significantly associated with patient characteristics, including age, sex, lipid-modifying therapy, and CV risk level. Authors concluded that in their country LDL-C goal achievement among high CV risk patients was suboptimal, with the lowest rate of LDL-C goal achievement in the higher CV risk patients. This is not an unusual finding, even if the large number of patients included and the well written manuscript, as well as the importance to have informations from particular ethnic groups like the South East Asians, represent the most noticeable points of strength of the manuscript. The study has also some limitations which are typical for retrospective clinical research and which were thoroughly discussed by the authors in the final section of their Discussion.

6. PLOS authors have the option to publish the peer review history of their article (what does this mean?). If published, this will include your full peer review and any attached files.

Reviewer #1: No

Reviewer #2: Yes: Alessandro Lupi

---

## [Author Response · Author response to Decision Letter 0]

19 Dec 2019

<Reviewer # 1>

1. What is the rate of CV events divided between LDL goal achievers and not-achievers considering the CV risk class?

[Authors’ response] We appreciate the Reviewer’s comment. To answer the Reviewer’s questions, the CV event rates between LDL goal achievers and not-achievers for each CV risk group were as follows: very high-risk group, 57.71/100 PYs and 45.82/100 PYs; high-risk group, 11.35/100 PYs and 11.17/100 PYs; moderate-risk group, 9.49/100 PYs and 9.26/100 PYs; low-risk group, 7.03/100 PYs and 6.72/100 PYs. The CV event rates were higher among achievers than non-achievers when patients were classified by CV risk level. In the previous version of manuscript which was submitted, CV event rates were not divided by CV risk groups, and the CV event rates were higher in non-achievers than in achievers (24.35/100 PYs vs 11.93/100 PYs; please refer to the lines 221-224 in the revised manuscript (clean version)). This is because the proportion of each CV risk group is different between achievers and non-achievers. As achievers consist of lower risk patients (very high-risk, 13.6%; high-risk, 22.4%; moderate-risk, 28.3%; low-risk, 35.7%) while non-achievers consist of higher risk patients (very high-risk, 57.7%; high-risk, 22.7%; moderate-risk, 12.7%; low-risk, 6.9%), summary rates across CV risk groups become low (i.e., more weighted by lower risk groups) in achievers (11.93/100 PYs) and become high (i.e., more weighted by higher risk groups) in non-achievers (24.35/100 PYs).

It should be noted that all of the CV event rates described above are the crude rates, which are not adjusted for differences in patients’ characteristics between achievers and non-achievers, and thus the results should be interpreted with caution. We presented the crude CV event rates to briefly show the characteristics of LDL-C goal achievers and non-achievers. As we mentioned in the Discussion section, precise comparisons of CV event rates between LDL-C goal achievers and non-achievers should be explored in future researches, by using appropriate statistical methods such as covariate adjustment or propensity score matching. Since the focus of this study was to examine the factors associated with LDL-C goal achievement, we did not include above information in the manuscript.

2. Cardiovascular events may have been influenced by failure to achieve targets in other CV risk factors (diabetes, hypertension). Do you have data on this?

[Authors’ response] Thank you for your comments. Unfortunately, we do not have the full data (e.g., HbA1c, etc.) to determine whether patients achieved their targets in other CV risk factors. The CV event rates we presented in the manuscript are the crude rates, which are not adjusted for other CV risk factors. As CV events are affected by various risk factors, including the achievement of diabetes and hypertension targets as the Reviewer’s comments, we are now planning the next research to estimate the CV event rates adjusted for various risk factors.

Although we do not have the full data to determine whether patients achieved their targets in other CV risk factors, we ran additional analyses using available data at best to check whether these CV risk factors influenced CV events. First of all, we have calculated crude CV event rates stratified by the achievement of diabetes and hypertension targets to figure out their effect on the CV event rates. The achievement of treatment targets of diabetes and hypertension was assessed on the index date, based on the fasting plasma glucose (<126 mg/dL) and systolic/diastolic blood pressures (<140/90 mmHg). The fasting plasma glucose was used instead of glycated hemoglobin, which was not available in the NHIS-HEALS database.

Among total patients, 58,939 (84%) achieved diabetes target and 54,684 (78%) achieved hypertension target. The CV event rates per 100 person-years stratified by the achievement of diabetes target, hypertension target, and LDL-C goal are as follows:

(i) Diabetes target achievers: 17.09 (all patients), 11.36 (LDL-C goal achievers), 24.28 (LDL-C goal non-achievers)

(ii) Diabetes target non-achievers: 20.60 (all patients), 15.89 (LDL-C goal achievers), 24.64 (LDL-C goal non-achievers)

(iii) Hypertension target achievers: 16.78 (all patients), 11.42 (LDL-C goal achievers), 23.41 (LDL-C goal non-achievers)

(iv) Hypertension target non-achievers: 20.78 (all patients), 14.07 (LDL-C goal achievers), 27.60 (LDL-C goal non-achievers)

For both diabetes and hypertension target, CV event rates were higher in non-achievers than achievers. However, the differences in CV event rates between achievers and non-achievers were much smaller for diabetes and hypertension target compared to that for LDL-C goal. Furthermore, LDL-C goal achievers showed consistently lower CV event rates than non-achievers regardless of the achievement of diabetes and hypertension target. As we mentioned in the comment #1, CV event rates were presented to briefly show the characteristics of LDL-C goal achievers/non-achievers and not the main focus of this study. Estimating CV event rates with consideration of various CV risk factors would be explored in future researches, by using full data which can evaluate the achievement of other CV risk factors. Thus, we did not include above information in the manuscript.

3. No data on BMI?

[Authors’ response] We appreciate the Reviewer for pointing this out. As we also thought that BMI could be an important factor associated with LDL-C goal achievement, we revised our logistic regression to consider BMI data. In the revised logistic regression, where 26 patients without BMI data were excluded, BMI showed significant association with LDL-C goal achievement (Please see Figure 5, lines 175-176, lines 240-244, lines 268-269, and lines 368-373 in the revised manuscript (clean version)). 

“…Patients without body mass index (BMI) data on the index date were excluded from the logistic regression. …”

“…Compared with patients with normal weight, underweight patients were more likely to achieve the goals (OR=1.53; 95% CI [1.28–1.82]) while overweight or obese patients were less likely to achieve the goals (OR=0.84; 95% CI [0.80–0.88] and OR=0.83; 95% CI [0.80–0.87] for overweight and obese, respectively). …”

“…Patients’ age, sex, LMT, CV risk level, BMI, disease severity, and HDL-C were associated with LDL-C goal achievement.”

“…Patients were less likely to achieve the LDL-C goal if they were female, overweight or obese, taking LMT other than statin, and with higher CV risk. More intensive LDL-C management, including regular LDL-C measurement and optimized lipid-modifying therapy, should be highlighted in patients who are less likely to achieve the LDL-C goal, such as female, overweight or obese patients, patients not adherent to statin, or patients with very high or high CV risk.”

Also we have added BMI data for the summary of baseline characteristics of study population (Please see Table 1 in the revised manuscript (clean version)). 

4. Are there data on chronic renal failure?

[Authors’ response] Thank you for your comments. Among the study population, only 721 (1.0%) had chronic renal failure and the proportion was not significantly different between LDL-C goal achievers and non-achievers (0.98% vs 1.07%, P-value=0.23). We have added this data for the summary of baseline characteristics of study population (Please see Table 1 in the revised manuscript (clean version)).

 

<Reviewer # 2>

1. In this retrospective cohort study the authors used data from a South Korean National Health Insurance DataBase (2006 to 2013) to assess the LDL-C goal achievement among patients with very high or high CV risk treated with statins and other lipid lowering drugs. The 2015 Korean guidelines were used to measure LDL-C goal achievement based on the CV risk level. 69,942 patients were evaluated. About 50% of the patients with dyslipidemia achieved their recommended LDL-C goal, but the achievement rates were substantially different across CV risk levels (17.6%, 47.2%, 66.9%, and 82.4% for very high-, high-, moderate-, and low-risk groups, respectively; P<0.0001). The crude event rate of total CV events during the follow-up period in the LDL-C goal non-achievers was higher than that in the LDL-C goal achievers (24.35/100 PYs vs. 11.93/100 PYs; P<0.0001). LDL-C goal achievement was significantly associated with patient characteristics, including age, sex, lipid-modifying therapy, and CV risk level. Authors concluded that in their country LDLC goal achievement among high CV risk patients was suboptimal, with the lowest rate of LDL-C goal achievement in the higher CV risk patients. This is not an unusual finding, even if the large number of patients included and the well written manuscript, as well as the importance to have informations from particular ethnic groups like the South East Asians, represent the most noticeable points of strength of the manuscript. The study has also some limitations which are typical for retrospective clinical research and which were thoroughly discussed by the authors in the final section of their Discussion.

[Authors’ response] We are grateful for the Reviewer’s comments. 

 

<Journal Requirements>

[Authors’ response] We checked and followed the PLOS ONE’s style requirements.

2. We note that you have indicated that data from this study are available upon request. PLOS only allows data to be available upon request if there are legal or ethical restrictions on sharing data publicly. For information on unacceptable data access restrictions, please see http://journals.plos.org/plosone/s/data-availability#loc-unacceptable-data-accessrestrictions. 

[Authors’ response] Thank you for the comments. We revised the Data Availability statement as per the comments. We included the statement in the cover letter.

“Data Availability: The datasets used and analyzed during the current study are not publicly available. There are legal or ethical restrictions on sharing this data publicly. Data for these analyses were made available to the authors through National Health Insurance Service (NHIS), through a formal application process (https://nhiss.nhis.or.kr/bd/ab/bdaba021eng.do). Researchers can access the NHIS data only when they meet the all of following conditions: (i) Korean citizenship; (ii) Institutional Review Board approval; and (iii) Permission by NHIS data provision review committee (contact number: +82-33-736-2431). Because NHIS strictly prohibits researchers from sharing the raw data (i.e., individual-level data which is not summarized) to unauthorized persons who are not involved in the study, other researchers who are interested in using NHIS database should request the data through a formal application process. For patient confidentiality reasons, public data sharing is restricted even if the data is anonymized.”

‘I have read the journal’s policy and the authors of this manuscript have the following competing interests: [Qian, Rane are employees of Amgen Inc and own stocks in the company. Zhao was an employee of Amgen Inc at the time this research was performed. Suh, S Kim, and Han received research grants from National Research Foundation, Ministry of Health and Welfare, Ministry of Food and Drug Safety, Korea Health Industry Development Institute, Abbvie Korea, Amgen Inc, Amgen Korea, Handok-Teva, Ipsen Korea, and Pfizer Korea.]'

We note that one or more of the authors are employed by a commercial company: Amgen Inc

[Authors’ response] Thank you for the comments. We revised the Financial Disclosure and Competing Interests statements as per the comments. We included both an updated Funding Statement and Competing Interests Statement in the cover letter.

“Financial Disclosure: This research was funded by Amgen, Inc.; URLs to sponsor’s website: https://www.amgen.com/. Qian, Rane are employees of Amgen Inc and own stocks in the company. Zhao was an employee of Amgen Inc at the time this research was performed. The funder provided support in the form of salaries for authors PPR, YQ, and ZZ, but the funder did not have any additional role in the study design, data collection and analysis, decision to publish, or preparation of the manuscript. The specific roles of these authors are articulated in the ‘author contributions’ section (Qian, Rane and Zhao contributed in development of study design, analysis, decision to publish and in preparation of the manuscript).”

“Competing Interests: I have read the journal’s policy and the authors of this manuscript have the following competing interests: [Qian, Rane are employees of Amgen Inc and own stocks in the company. Zhao was an employee of Amgen Inc at the time this research was performed. Suh, S Kim, and Han received research grants from National Research Foundation, Ministry of Health and Welfare, Ministry of Food and Drug Safety, Korea Health Industry Development Institute, Abbvie Korea, Amgen Inc, Amgen Korea, Handok-Teva, Ipsen Korea, and Pfizer Korea.] This does not alter our adherence to PLOS ONE policies on sharing data and materials.”

---

## [Editor Report · Decision Letter 1]

16 Jan 2020

Achievement of the low-density lipoprotein cholesterol goal among patients with dyslipidemia in South Korea

PONE-D-19-24724R1

Dear Authors

We are pleased to inform you that your manuscript has been judged scientifically suitable for publication and will be formally accepted for publication once it complies with all outstanding technical requirements.

With kind regards,

Luigina Guasti

Academic Editor

PLOS ONE
---

## [Editor Report · Acceptance letter]

22 Jan 2020

PONE-D-19-24724R1 

Achievement of the low-density lipoprotein cholesterol goal among patients with dyslipidemia in South Korea 

Dear Dr. Suh:

I am pleased to inform you that your manuscript has been deemed suitable for publication in PLOS ONE. Congratulations! Your manuscript is now with our production department. 

With kind regards,

on behalf of

Dr. Luigina Guasti 

Academic Editor

PLOS ONE